# Fatty Acid Profile of Mature Red Blood Cell Membranes and Dietary Intake as a New Approach to Characterize Children with Overweight and Obesity

**DOI:** 10.3390/nu12113446

**Published:** 2020-11-10

**Authors:** Iker Jauregibeitia, Kevin Portune, Itxaso Rica, Itziar Tueros, Olaia Velasco, Gema Grau, Nerea Trebolazabala, Luis Castaño, Anna Vita Larocca, Carla Ferreri, Sara Arranz

**Affiliations:** 1AZTI, Food Research, Basque Research and Technology Alliance (BRTA), Parque Tecnológico de Bizkaia, Astondo Bidea, Edificio 609, 48160 Derio, Spain; ijauregibeitia@azti.es (I.J.); kportune@azti.es (K.P.); itueros@azti.es (I.T.); 2Biocruces Bizkaia Health Research Institute, Cruces University Hospital, CIBERDEM/CIBERER, UPV/EHU, 48903 Barakaldo, Spain; itxaso.ricaechevarria@osakidetza.eus (I.R.); olaia.velascovielba@osakidetza.eus (O.V.); MARIAGEMA.GRAUBOLADO@osakidetza.eus (G.G.); nerea.trebolazabalaquirante@osakidetza.eus (N.T.); luisantonio.castanogonzalez@osakidetza.eus (L.C.); 3Lipidomic Laboratory, Lipinutragen srl, Via di Corticella 181/4, 40128 Bologna, Italy; annavita.larocca@lipinutragen.it; 4Consiglio Nazionale delle Ricerche, ISOF, Via Piero Gobetti 101, 40129 Bologna, Italy

**Keywords:** childhood obesity, inflammation, membrane lipidome, omega-6 fatty acids, red blood cell

## Abstract

Obesity is a chronic metabolic disease of high complexity and of multifactorial origin. Understanding the effects of nutrition on childhood obesity metabolism remains a challenge. The aim of this study was to determine the fatty acid (FA) profile of red blood cell (RBC) membranes as a comprehensive biomarker of children’s obesity metabolism, together with the evaluation of their dietary intake. An observational study was carried out on 209 children (107 healthy controls, 41 who were overweight and 61 with obesity) between 6 and 16 years of age. Mature RBC membrane phospholipids were analyzed for FA composition by gas chromatography-mass spectrometry (GC-MS). Dietary habits were evaluated using validated food frequency questionnaires (FFQ) and the Mediterranean Diet Quality Index for children (KIDMED) test. Compared to children with normal weight, children with obesity showed an inflammatory profile in mature RBC FAs, evidenced by higher levels of ω-6 polyunsaturated FAs (mainly arachidonic acid, *p* < 0.001). Children who were overweight or obese presented lower levels of monounsaturated FA (MUFA) compared to children with normal weight (*p* = 0.001 and *p* = 0.03, respectively), resulting in an increased saturated fatty acid (SFA)/MUFA ratio. A lower intake of nuts was observed for children with obesity. A comprehensive membrane lipidomic profile approach in children with obesity will contribute to a better understanding of the metabolic differences present in these individuals.

## 1. Introduction

Interventions to control obesity have typically consisted of combined strategies including diet, exercise, and behavior therapy [1]. Despite efforts by governments, the food industry, and the science community, obesity and overweight rates keep increasing in both child and adult populations worldwide [2,3], demonstrating a need for personalized strategies that guarantee the success of interventions in treating obesity.

The focus on early clinical markers for overweight/obesity onset is, nowadays, a clear research target [4]. A new trend to focus on fats, and specifically the quality of dietary lipids, is crucial for the prevention and treatment of obesity [5]. In this sense, a strong contribution from the molecular approach developed in the last two decades, characterizing fat accumulation, highlights different kinds of signaling occurring in this disease and leading to comorbidities [6]. Since fat accumulation is strictly connected with the quality and quantity of fatty acids (FAs) in human tissues, the lipidomic approach was found to have a key role in describing the scenario of molecular signaling, providing crucial information on the various phases of weight increase, from overweight to obesity [7].

Indeed, membrane fatty acid-based lipidomics has reached a high technology readiness level, developing simple (i.e., inexpensive and high-throughput) and robust analytics of high resolving power, as demonstrated by several applications to diseases [8,9].

Molecular information on membrane lipid composition is of great importance at least for two reasons: (i) the set of membrane fatty acid controls the fluidity and permeability properties as well as the thickness of the bilayer, which are all implicated in the receptor and channel responses [10], and (ii) lipid signaling departs from the fatty acid residues of the membrane phospholipids. Therefore, the fatty acid composition directly describes the cellular predisposition to respond to the various stimuli that arrive from the extracellular environment [11]. Therefore, the balance in the fatty acid composition of the cell membrane leads to the balance in the functions of each individual cell and, hence, of the tissues and the whole organism [12].

It is worth recalling that to measure the lipid composition of blood, different blood compartments have been targeted [13]. Plasma or serum FA levels have been widely analyzed because they reflect short-term dietary fat intake. However, analysis of lipid compositions from mature red blood cell (RBC) membranes offers an advantage over analysis of plasma because these cells last on average 120 days in the blood compared to 3 weeks for platelet or plasma lipids, reflecting better long-term dietary FA intake and tissue conditions [14]. Apart from this, RBCs maintain a more stable FA composition compared to plasma FA levels [15].

It is important to note that although future nutritional intervention studies are necessary to better understand the impact of personalized diet on lipid metabolism in children, lipidomics can help monitor the ω-6 fatty acid content involved in the inflammation pathways that can be accompanied by essential FA deficiencies in the diet, which can be connected to many diseases and tissue malfunctions. As a matter of fact, monitoring the RBC membrane FA profile at the individual level can be an excellent candidate biomarker as it can offer the possibility to follow up the optimal intake, membrane incorporation, and biochemical transformations in order to personalize dietary intervention designed to recover FA deficiencies to prevent or control disease. Fatty acids in phospholipids represent the combination between nutritional and metabolic factors, with a strong contribution of the individual metabolism and condition of the patients.

This study aims to generate knowledge on the importance of different fatty acid families (saturated fatty acids (SFA), monounsaturated fatty acids (MUFA), and polyunsaturated fatty acids (PUFA)) for the cell membrane lipidome in a pediatric population. In this exploratory study, analysis of the RBC membrane shows its potential to provide indications of dietary and metabolic distinctions between children with overweight and obesity that can contribute, in the future, to designing more precise nutritional strategies that may be more effective at correcting the molecular imbalance in obesity.

## 2. Materials and Methods

### 2.1. Study Design

An observational case-control and retrospective study was conducted on 209 children (113 boys and 96 girls) between 6 and 16 years old, recruited from the pediatric endocrinology unit at the Hospital Universitario Cruces (Barakaldo, Spain). Children were classified according to body mass index (BMI), using an age and sex-specific pediatric z-score from Orbegozo tables [16]. BMI was taken as a reference to define the different categories, defining normal weight when the standard deviation (SD) of BMI was −1 < SD ≤ +1, overweight when the +1 < SD ≤ +2, and obesity when SD > +2. Groups were homogeneously distributed by gender and age. Finally, 107 children with normal weight, 41 children with overweight, and 61 children with obesity were enrolled in the study.

Subjects were excluded if they presented any kind of acute or chronic diseases, were taking medications, or had any presence of metabolic syndrome symptoms or obesity associated to any type of pathology. A physical examination was performed by a pediatrician.

The study protocol was approved by the Euskadi Clinical Research Ethics Committee (permission number PI2016181) and accomplished according to the Helsinki Declaration in 1975, revised in 2013. Subjects under study were included after acceptance (of the parents) to participate in the study and signing of informed consent. All the informed consent documents were signed by their parents, and in the case of children between 12 and 16 years of age, the informed consent was also signed by themselves, according to the Euskadi Ethical Committee and sample biobank laws (Organic Law 3/2018, of December 5, on Protection of Personal Data and guarantee of digital rights; Law 14/2007 on Biomedical Research and RD 1716/2011 of Biobanks).

### 2.2. Anthropometric Measures

Body weight (kg) and height (cm) were measured by standardized methods [17]. Body mass index (BMI) was calculated as weight (kg) divided by the square of the height (m^2^). Anthropometric parameters as well as blood sampling were all conducted by pediatricians during the first visit to the Hospital Universitario Cruces.

### 2.3. Food Habits and Nutrient Intakes

During the first visit, a pediatrician interviewed the participants and collected personal data, including family medical history and information on the history of medication usage. Estimations of food consumption, including dietary diversity and variety, were measured using a quantitative food frequency questionnaire (FFQ), completed online by the parents of each volunteer, except in those cases of adolescents, who were encouraged to complete it themselves. For our study, an adapted FFQ was used, which was previously validated with portion sizes and food groups for the Spanish juvenile population [18,19,20]. Information about different food items collected from these questionnaires was then analyzed using DIAL^®^ software to translate their intake into their corresponding energy and nutrient composition (UCM & Alce Ingeniería S.A, Madrid, Spain) (v3.4.0.10).

Dietary habits were also measured using the KIDMED test (Mediterranean Diet Quality Index), a validated questionnaire for the Spanish juvenile population that measures adherence to the Mediterranean diet, which is widely considered to be an optimally healthy diet for most populations [21,22]. According to the KIDMED index, a score of 0–3 reflects poor adherence to the Mediterranean diet, a score of 4–7 describes average adherence, and a score of 8–12 good adherence.

### 2.4. Red Blood Cell (RBC) Membrane Fatty Acid Analysis

The fatty acid composition of mature RBC membrane phospholipids was obtained from blood samples (approximately 2 mL) collected in vacutainer tubes containing ethylenediaminetetraacetic acid (EDTA). Samples were shipped to the Lipidomic Laboratory at a controlled temperature and, upon arrival, underwent quality control for the absence of hemolysis. During the blood work-up, before lipid extraction and lipid transesterification to fatty acid methyl esters (FAMEs), the automated protocol includes the selection of mature RBCs, as reported previously [9,23,24,25]. Briefly, the whole blood in EDTA was centrifuged (4000 revolutions per minute (rpm) for 5 min at 4 °C) and the mature cell fraction was isolated based on the higher density of the aged cells [26] and controlled by the use of a cell counter (Scepter 2.0 with Scepter™ Software Pro, EMD Millipore, Darmstadt, Germany). All the subsequent steps were automated and included cell lysis, isolation of the membrane pellets, phospholipid extraction from pellets using the Bligh and Dyer method [27], transesterification to FAMEs by treatment with a potassium hydroxide (KOH)/methyl alcohol (MeOH) solution (0.5 mol/L) for 10 min at room temperature, and extraction using hexane (2 mL). The FAMEs were analyzed using capillary column gas chromatography (GC). GC analysis was run on the Agilent 6850 Network GC System (Agilent, USA), equipped with a fused silica capillary column Agilent DB23 (60 m × 0.25 mm × 0.25 μm) and a flame ionization detector. Optimal separation of all fatty acids and their geometrical and positional isomers was achieved. Identification of each fatty acid was made by comparison to commercially available standards and to a library of trans isomers of MUFAs and PUFAs. The amount of each FA was calculated as a percentage of the total FA content (relative %), as described in Section 2.5, being more than 97% of the GC peaks recognized with appropriate standards.

### 2.5. Red Blood Cell Membrane Fatty Acid Cluster

Twelve FAs were chosen as representative cluster of the main building blocks of the RBC membrane glycerophospholipids and of the three FA families (SFA, MUFA, and PUFA): for SFAs, palmitic acid (C16:0) and stearic acid (C18:0); for MUFAs, palmitoleic acid (C16:1; c9), and oleic acid (C18:1; c9), cis-vaccenic acid (C18:1; c11); for ω-3 PUFAs, eicosapentaenoic acid (EPA) (C20:5) and docosahexaenoic acid (DHA) (C22:6); for ω-6 PUFAs, linoleic acid (LA) (C18:2), dihomo-gamma-linolenic acid (DGLA) (C20:3), and arachidonic acid (AA) (C20:4); for geometrical trans fatty acids (TFAs): elaidic acid (C18:1 t9) and mono-trans arachidonic acid isomers (monotrans-C20:4; ω-6 recognized by standard references as previously described by Ferreri et al. [28]). Considering these fatty acids, different indexes previously reported in the literature [25] were calculated: Omega-3 Index: (%EPA + %DHA) an index suggested as a cardiovascular disease risk factor; (%SFA/%MUFA) index related with membrane rigidity; inflammatory risk index (% ω-6)/(% ω-3); PUFA balance (%EPA + %DHA)/total PUFA × 100; free radical stress index (sum of trans-18:1 + summary (Σ) of monotrans 20:4 isomers); unsaturation index (UI) (%MUFA) + (%LA/2) + (%DGLA/3) + (%AA/4) + (% EPA/5) + (%DHA/6); peroxidation index (PI) (%MUFA/0.025) + (%LA) + (%DGLA/2) + (%AA/4) + (% EPA/6) + (%DHA/8).

Additionally, the enzymatic indexes of elongase and desaturase enzymes, the two classes of enzymes of the MUFA and PUFA biosynthetic pathways, were inferred by calculating the product/precursor ratio of the involved FAs.

### 2.6. Statistical Analysis

A power calculation was performed using G*power software (v3.1.9.7., Heinrich-Heine-University, Düsseldorf, Germany), to determine the total sample size for analysis of covariance (ANCOVA) fixed effects, main effects, and interactions. A priori, we expected a medium effect size f = 0.25, (as the ratio of the variation among the group means to the average variation among subjects within each group as measured by their standard deviations), using an alpha = 0.05 as probability of the type I error, to have a 95% confidence for significative results, and beta = 0.2, as the acceptable probability of type II errors, concluding in a 0.8 power (where power is equal to 1-beta). It is estimated that a lower value would imply too great a risk of incurring a type II error. A higher value would imply excessively expanding the sample [29]. The total sample size required was 158, which corresponds with 53 participants for each group.

Differences between groups for nutrient intake, food group intake, and KIDMED test were determined by conducting a Kruskal-Wallis test because the data were not normally distributed. Normal data distribution was assessed by Shapiro-Wilk’s test or/and the Kolmogorov-Smirnov test. Subsequently, Dunn’s (1964) test was performed for post hoc comparisons. A Bonferroni correction for multiple comparisons was made, to correct for the increased risk of type I error.

An analysis of covariance test (ANCOVA) was run to determine the differences between RBC membrane fatty acids from children with normal weight, overweight, and obesity, after controlling for variables selected as potential confounders, such as age, gender, and dietary macro- and micronutrient intake. Post hoc analysis was performed with a Bonferroni adjustment for multiple comparisons. First, a principal component analysis (PCA) was run on 18 dietary nutrient intake variables (individual FAs, families (SFA, MUFA, and PUFA), total lipids (%Energy), carbohydrates, fiber, proteins, and calories), obtained with DIAL software (v3.4.0.10, Department of Nutrition (UCM) & Alce Ingeniería, S.L., Madrid, Spain) after transforming the information about food items from FFQ questionnaires into micro- and macronutrient values, in order to reduce and simplify the dimensions of these variables and use the generated factors as diet covariates [25]. Kaiser-Meyer-Olkin (KMO) and Bartlett’s test of sphericity were used to verify the sampling adequacy for the analysis. The PCA revealed three components that had eigenvalues greater than one and which explained 83.74% of the total variance. These components were included in the ANCOVA analysis as diet covariates. The level of significance was set at *p* < 0.05.

In order to establish correlations between the RBC membrane FA profile and the dietary intake and other parameters measured in this study, due to the non-normality of the data of most of the variables, Spearman’s rank-order correlation was conducted. In those cases where all variables had a normal distribution, a Pearson product-moment correlation was run. All statistical analyses were performed using SPSS (IBM Corp. v24.0, Armonk, NY, USA).

## 3. Results

### 3.1. Dietary Intake

Table 1 shows dietary intake according to food categories calculated via food questionnaires. The diet of children with obesity was characterized by lower intake of cereals (*p* = 0.04), dairy products (*p* = 0.05), and nuts (*p* = 0.01), compared to children with normal weight. The overweight group only showed significant differences of lower intake of cereals compared with the group of children with normal weight (*p* = 0.004). Regarding diet quality, the KIDMED questionnaire was conducted to measure the adherence of different groups to the Mediterranean diet. As we can see in Table 1, only children with normal weight achieved good adherence (KIDMED score ≥8), while children with overweight and obesity showed mild adherence to the Mediterranean diet (less than 8 points and equal to or greater than 4 points) [21]. Even so, children with normal weight just had a difference of one point in the KIDMED scale compared with the other groups and no statistically significant differences were observed, (*p* = 0.07). KIDMED and BMI showed a statistically significant slight negative correlation (Pearson’s Correlation coefficient r_s_ (98) = -0.198, *p* = 0.004). The KIDMED index did show some slight correlation with RBC membrane AA, EPA, and DHA levels (r_s_ (98) = −0.183, *p* = 0.010, r_s_ (98) = 0.195, *p* = 0.006, and r_s_ (98) = 0.227, *p* = 0.001, respectively).

Table 2 shows the differences among three groups in macronutrients and individual fatty acid daily intake expressed as % of Kcal. No differences among groups were observed for macronutrients, except for the group of children with normal weight that showed a lower intake of total lipids compared with the overweight group (*p* = 0.01). Considering the intake of specific fatty acids, the children with normal weight reported a higher intake of C16:0 and total SFAs compared to the groups with overweight and obesity, but no other differences were observed.

### 3.2. RBC Membrane Fatty Acid Profile

In order to compare RBC FA profiles between groups, a one-way ANCOVA was conducted using age, sex, and dietary intake as covariates to adjust the error made by those confounding factors (Table 3). The group with obesity showed higher levels of stearic acid (*p* = 0.03) and total SFA (*p* = 0.03) than the normal weight group. Oleic acid and total MUFA levels in the groups with obesity and overweight were lower compared with the children with normal weight. Regarding *ω*-6 FA, linoleic acid was higher for the normal weight group compared to children with obesity (*p* = 0.03), but dihomo-γ-linolenic acid (DGLA) and arachidonic acid levels were higher for children with obesity compared with the children with normal weight (*p* = 0.002 and *p* = 0.0003, respectively). Individual and total *ω*-3 levels did not show significant differences. The SFA/MUFA ratio was higher for children with obesity and overweight compared with children with normal weight (*p* = 0.001 and *p* = 0.03, respectively), and for the *ω*-6/*ω*-3 ratio, children with obesity and overweight showed higher but not statistically significant values compared with the children with normal weight (*p* = 0.09 and *p* = 0.1, respectively).

With respect to enzymatic activity, ∆9D was lower for children with obesity, indicating the hypoactivity of this enzyme to convert stearic acid (C18:0) to oleic acid (9c C18:1).

In order to explore possible relationships between dietary intake components and RBC lipid profile, a Spearman’s rank-order correlation was carried out. Only dietary EPA, DPA, and DHA showed mild correlations with levels of EPA in the RBC (r_s_ (98) = 0.33 *p* < 0.0001, r_s_ (98) = 0.363 *p* < 0.0001, and r_s_ (98) = 0.313 *p* < 0.0001, respectively).

At the same time, dairy products and cereal intake were higher for children with normal weight compared to children with obesity, but neither showed significant correlations with the RBC membrane FAs.

## 4. Discussion

Lipidomic monitoring of mature RBC membranes evaluated in this study contributes to highlighting the importance of each fatty acid class as a molecular parameter to better understand the lipidomic pathways connected with childhood obesity. Our results indicate molecular inflammation derived from unbalanced levels of membrane fatty acids and dysregulation of desaturase enzymatic activity as key parameters for the metabolic outcome in obesity.

Different studies have been published reporting RBC membrane lipid profiles of adult populations with overweight and obesity [30,31] and a few, also, on child populations with obesity [32].

In a meta-analysis study published by Fekete et al. [33], the FA profiles from different blood fractions were analyzed in order to determine the long-chain PUFA status in obesity. In total plasma lipids and phospholipids, high variability in individual FA levels was observed throughout the different studies analyzed. All the studies agreed that differentiated lipid profiles were observed for subjects with overweight and obesity compared to subjects with normal weight, characterized by a greater alteration in ω-6 FA. In agreement with our results, all the biomarkers analyzed in this meta-analysis showed increased levels of DGLA and decreased LA for the population with overweight and/or obesity. This meta-analysis could not find any significant result for AA, despite increased levels of AA in adipose tissue, which has been previously associated with obesity [34,35,36]. However, our results revealed significant increased levels of AA in RBC membranes in children with obesity. This heterogeneity points towards a need for a more precise biomarker to characterize and compare different population groups, highlighting the advantages of choosing mature RBC membrane as a representative of nutritional and metabolic contributions [11].

Our results can be seen also in view of other studies that have also analyzed RBC membranes. A study on children with overweight and obesity with metabolic syndrome from 5 to 18 years of age in an Italian population showed similar FA levels of total ω-6 and total SFAs, whereas for all the other measured PUFAs and indexes, differences could be observed [32]. The variability in RBC FA between different countries [37] might be the reason, and this is an important point to emphasize in our approach, which proposes common features of an automatized procedure for cell sampling and membrane isolation and of a precise cluster of fatty acids to analyze.

To our knowledge, this is the first time that a systematic approach was employed that analyzes FAs from isolated mature RBC membranes in pediatric populations with overweight and obesity compared to children with normal weight in order to identify specific characteristics of the fatty acid profile for childhood obesity. Furthermore, as the composition of the RBC membrane is substantially affected by diet and metabolism, the elimination of the effect of diet as a confounding variable in our ANCOVA analysis allows a more robust and realistic examination of the effects of metabolic status of the obesity condition on the RBC FA profile.

According to data obtained in our study, the group with obesity is characterized by an increase in ω-6 fatty acids due to the higher levels of AA and, at the same time, of DGLA, but ω-3 mediated signaling also has to be balanced. Omega-6 FAs have been previously described in the literature as precursors of proinflammatory mediators [38,39] that act through different mechanisms on inflammatory processes. Unlike the other ω-6 FAs, LA showed lower levels for the group with obesity compared with the children with normal weight. Higher RBC LA levels have been linked with improved body composition, insulin resistance, and lower levels of inflammatory markers in previous studies. This disequilibrium of PUFA metabolism towards ω-6 FAs seems to contribute to excessive adipose tissue development and represents, itself, an emerging risk factor for obesity [40,41].

Previous studies have shown an inverse correlation between ω-3 intake and AA levels in RBCs that can be due the competition of the Δ6-desaturase [42]. Although we did not observe differences in the dietary intake of ω-3, nor in RBC ω-3 levels, but considering that AA levels appear to be enhanced in obesity, an increase in ω-3 consumption in the population with obesity can be a crucial dietary recommendation in order to counteract proinflammatory precursors linked with the disease.

Regarding SFA and MUFA levels, an altered ratio in the group with obesity can be observed, mainly due to lower levels of oleic acid and higher levels of stearic acid in the group with obesity compared with the children with normal weight. The enzymatic activity of Δ-9-desaturase or stearoyl-CoA desaturase-1 (SCD1), measured indirectly by the ratio between oleic and stearic acids, showed a lower activity for the group with obesity. The overall picture that comes from examination of the fatty acid remodeling occurring in obesity highlights the role of de novo lipogenesis with the formation of saturated fatty acids (SFA) and their enzymatic transformation to monounsaturated fatty acids (MUFA), connected with the functioning of delta-9 desaturase and the corresponding gene expression (SCD1, Stearoyl CoA Desaturase). As a consequence, the main fatty acid biomarkers of weight increase are MUFAs, such palmitoleic acid (9 cis-16:1) and oleic acid (9 cis-18:1), the former being considered for its role as a lipokine [43] and the latter being considered the main fatty acid accumulating in adipose tissue as triglycerides [44,45,46,47,48].

Although higher dietary intake of SFAs was observed in the normal weight group compared with the group with obesity, the SFA level in RBC membranes was slightly higher in the study population with obesity. Possible explanations for this result, as explained above, could be due to the greater activity of SCD1 in the children with normal weight, which may have converted higher proportions of SFAs to MUFAs, as reflected in higher oleic acid levels in this group. At the same time, the SFA/MUFA ratio, which is correlated with increased membrane rigidity, appears in higher levels in the group with obesity [12,49].

Precision nutrition based on molecular data considers that the assessment of dietary patterns provides a more reliable picture of real food intake compared to the assessment of individual macronutrient intake [50]. Links between dietary patterns and RBC lipid composition have been considered in our study to provide information that could be useful for more precise nutritional recommendations. Dietary patterns of children with normal weight were characterized by higher intake of nuts compared with the group with obesity and a higher intake of cereals compared with the overweight group, but neither food groups showed correlations with the RBC membrane FAs. However, different epidemiological and nutritional clinical trials conducted in adults have reported an inverse relationship between nut consumption and body mass index (BMI) [51,52,53], associated with several health benefits, such as antioxidant, hypocholesterolemic, cardioprotective, anticancer, anti-inflammatory, and antidiabetic benefits, among other functional properties [54,55]. Previously published results based on self-reported intakes using food frequency questionnaires pointed out that a high intake of grains could be protective against obesity [56,57]. Regarding diet quality, measured by the KIDMED test, statistically significant differences were not observed. Even so, a tendency of a higher adherence to Mediterranean diet in the children with normal weight, compared with the groups with overweight and obesity, can be observed (*p =* 0.07), and these results agree with other studies with pediatric populations with overweight and obesity [58].

In our opinion, there are several points to be considered for the identification of a fat biomarker in obesity, as well as in other physio-pathological states. The first focus is on the choice of the sample to examine: (i) mature red blood cells are representative for both functional and structural roles of the fatty acid residues which compose its membrane phospholipids; (ii) the fatty acids in phospholipids represent the combination between nutritional and metabolic factors, with strong contribution to the individual metabolism and condition of the patients.

The second focus is on the variations in the fatty acid residues of membrane phospholipids, which point to a differentiation between the overweight and obese status: (a) oleic acid is reduced in both subjects with overweight and those with obesity, which is interesting since, in adults, it is known that the decrease in this fatty acid is correlated with weight increase [31]. The role of the FFQ checked in our study clarified that there is no increase in SFA intake in children with obesity, thus shifting the attention to the metabolism of children with obesity and enzymatic functioning. It should be noted that in overweight and obesity, the decrease in desaturase delta-9 activity is significant, thus suggesting a shift in the enzyme functioning toward the SFA pathway, and (b) the role of the omega-6 pathway for the arachidonic acid increase is clearly shown in the comparison between children with normal weight and children with obesity, showing that in obesity, the increase in inflammatory signals could be crucial to be controlled in order to treat metabolic deficiencies related to childhood obesity.

This is an exploratory study that highlights some aspects to be further assessed in larger populations but also gives an indication for dietary and metabolic distinctions between children with overweight and obesity, to be further explored with intervention studies. At the same time, the indirect measurement of enzyme activity by the ratio between product and precursors, although very popular, could be considered as a limitation of the study and should be measured directly to emphasize and reaffirm the conclusions obtained. The use of an FFQ is also a limitation of the study as this type of questionnaire usually underestimates dietary intake [59]. Regarding the sample size, the authors are aware that the overweight group did not achieve the expected sample size, which was attributed to difficulties in recruiting, typically observed in human studies and especially in recruiting children; therefore, there might not have been enough power to detect differences in the overweight group and this could have led to a type II error. Future work should be focused on increasing the study sample size to achieve a greater statistical power. In order to compare with other studies that use international tables for determining obesity, such as the International Obesity Task Force (IOTF) or the World Health Organization (WHO), using the Spanish tables described above can be seen as a limitation, even if they better describe our study population. We can summarize that the presented study showed an altered RBC FA profile of the pediatric population with obesity, characterized by two main features: (i) an increased ω-6 molecular contribution, although no differences in PUFA dietary intake between groups were observed, which can create a predisposition for unbalanced signaling that departs from membranes; (ii) higher SFAs in the RBC membranes of children with obesity, contrary to the SFA intake that was found to be higher in controls, which can perturb membrane organization. Both results highlight the crucial role of molecular diagnostics for precise evaluation of patient status. Indeed, the lipidomic analysis of mature RBCs provides a systematic, automatized approach for the characterization of the lipid composition in these cells from the pediatric population with obesity, which can provide molecular insights to assist further development of precise and personalized nutritional strategies. Restoration of the optimal levels of each individual fatty acid, families, and ratios appears to be an important strategy to be considered in the treatment of potentially metabolic deficiencies related to childhood obesity from a nutritional point of view. Future studies, mainly nutritional intervention studies with children, are needed to elucidate an adequate selection among the types of fats that must be ingested, with the crucial target of the recovery of the homeostatic levels of the cell membrane, for proper functioning, both of signaling and metabolic pathways.

## Figures and Tables

**Table 1 nutrients-12-03446-t001:** Food group intakes.

Food Groups (g/day)	Children with Normal Weight (NO) *n* = 107	Children Who Are Overweight (OV) *n* = 41	Children with Obesity (OB) *n* = 61	Kruskal-Wallis H Test (*p*)	Post hoc Pairwise Comparison (*p* *)		
	Med (Q1–Q3)	Med (Q1–Q3)	Med (Q1–Q3)		NO:OV	NO:OB	OV:OB
Fruits	423 (297–532)	354 (273–509)	419 (272–603)	0.50			
Vegetables	159 (96–259)	154 (77–250)	141 (74–232)	0.49			
Cereals	161 (118–210)	127 (96–171)	139 (105–188)	0.01	0.01	0.11	0.95
Legumes	91 (54–102)	80 (51–102)	75 (48–96)	0.52			
Olive oil	15 (15–37)	15 (12–15)	15 (12–37)	0.26			
Dairy products	325 (255–512)	314 (226–329)	302 (207–358)	0.10			
Eggs	15 (15–34)	15 (15–34)	15 (15–34)	0.36			
Red meat	21 (21–21)	21 (21–50)	21 (21–50)	0.31			
White meat	50 (21−50)	50 (36–50)	50 (21–50)	0.15			
Dried fruits and nuts	2.1 (0–6.4)	1.1 (0–6.4)	1.1 (0–2.1)	0.03	0.73	0.03	0.96
Lean fish	27 (27–27)	27 (27–27)	27 (13–27)	0.613			
Oily fish and shellfish	27 (13–27)	27 (9–31)	27 (11–31)	0.55			
Sugary drinks	18 (0–45)	21 (0–54)	16 (0–54)	0.95			
Juices	80 (27–250)	107 (27–196)	80 (27–250)	0.95			
KIDMED score	8 (7–9)	7 (6–9)	7 (6–9)	0.07			

Data are expressed as medians and quartile 1 and quartile 3 (Med Q1-Q3). Not normally distributed variables. * Pairwise comparison conducted with a Bonferroni adjustment.

**Table 2 nutrients-12-03446-t002:** Dietary daily intake expressed as % of energy (%E).

Variables	Children with Normal Weight (NO) *n* = 107	Children Who Are Overweight (OV) *n* = 41	Children with Obesity (OB) *n* = 61	Kruskal-Wallis H Test (*p*)	Post Hoc Pairwise Comparison (*p* *)		
	Med (Q1–Q3)	Med (Q1–Q3)	Med (Q1–Q3)		NO:OV	NO:OB	OV:OB
Macronutrients							
Calories (Kcal/day)	2058 (1749–2376)	1983 (1516–2335)	1916 (1709–2167)	0.18			
Proteins (%E)	16.3 (15.0–17.7)	16.8 (15.3–18.5)	16.4 (14.9–17.3)	0.42			
Carbohidrates (%E)	46.7 (43.2- 49.9)	48.1 (43.9–53.3)	46.7 (43.4–51.1)	0.19			
Simple sugars (%E)	20.9 (18.5–23.8)	20.9 (17.9–24.8)	21.8 (18.9–25.0)	0.42			
Lipids (%E)	33.7 (29.9–37.0)	31.1 (27.0–35.4)	32.9 (27.4–38.2)	0.05			
Individual FA (% E)							
C14:0	1.0 (0.8–1.2)	0.8 (0.6–1.1)	0.9 (0.7–1.1)	0.02	0.06	0.08	1.0
C16:0	6.3 (5.7–7.1)	5.8 (5.1–6.7)	5.8 (5.2–6.7)	0.01	0.04	0.03	1.0
C18:0	2.4 (2.1–2.7)	2.2 (1.9–2.5)	2.2 (2.0–2.6)	0.05			
Tot. SFA	9.7 (8.7–10.9)	9.0 (7.8–10.1)	9.0 (7.9–10.4)	0.004	0.02	0.04	1.0
C16:1	0.51 (0.46–0.58)	0.49 (0.42–0.54)	0.50 (0.42–0.60)	0.1			
C18:1	14.2 (11.4–16.5)	12.3 (10.1–14.8)	13.3 (11.0–17.3)	0.08			
Tot. MUFA	14.7 (11.9–17.1)	12.7 (10.5–15.2)	13.9 (11.4–17.8)	0.07			
C18:2	3.6 (3.2–4.2)	3.6 (3.1–3.9)	3.6 (2.9–4.3)	0.99			
C20:4	0.04 (0.03–0.05)	0.04 (0.03–0.06)	0.04 (0.03–0.05)	0.78			
Tot. ω6	3.5 (3.0–4.4)	3.7 (2.8–4.5)	3.4 (3.1–5.0)	0.97			
>C18:3	0.52 (0.50–0.61)	0.50 (0.46–0.54)	0.52 (0.45–0.58)	0.44			
C20:5 (EPA)	0.07 (0.04–0.1)	0.07 (0.02–0.11)	0.07 (0.04–0.1)	0.88			
C22:5 (DPA)	0.017 (0.011–0.024)	0.017 (0.006–0.025)	0.016 (0.009–0.025)	0.59			
22:6 (DHA)	0.14 (0.09–0.19)	0.13 (0.05–0.20)	0.13 (0.09–0.19)	0.84			
Tot. ω3	0.8 (0.7–1.0)	0.8 (0.6–0.9)	0.8 (0.6–0.9)	0.36			
Tot. PUFA	4.3 (3.8–5.3)	4.5 (3.5–5.4)	4.3 (3.6–5.5)	0.96			
ω-6/ω-3	4.6 (4.0–5.4)	4.8 (3.7–6.8)	4.9 (4.0–6.7)	0.25			

Data are expressed as medians and quartile 1 and quartile 3 (Med Q1-Q3. Not normally distributed variables. SFA-saturated fatty acid; MUFA-monounsaturated fatty acid; PUFA-polyunsaturated fatty acid, (%E)-%Energy. * Pairwise comparison conducted with a Bonferroni adjustment.

**Table 3 nutrients-12-03446-t003:** Red blood cell (RBC) membrane fatty acid profile.

Fatty Acids (%)	Children with Normal Weight (NO)		Children Who Are Overweight (OV)			>Children with Obesity (OB)	ANCOVA	*p*-Value ^a^		
	Mean	SE	Mean	SE	Mean	SE	*p*	NO:OV	NO:OB	OV:OB
Palmitic acid (C16:0)	22.44	0.10	22.54	0.16	22.49	0.13	0.86	1.00	1.00	1.00
Stearic acid (C18:0)	17.67	0.10	17.94	0.17	18.13	0.14	0.03	0.54	0.03	1.00
TOT. SFA	40.12	0.10	40.48	0.16	40.58	0.14	0.02	0.21	0.03	1.00
Palmitoleic acid (C16:1)	0.40	0.01	0.45	0.02	0.43	0.02	0.08	0.12	0.31	1.00
Oleic acid (9c C18:1)	17.48	0.13	16.68	0.20	16.65	0.17	<0.001	<0.001	<0.001	1.00
cis-Vaccenic acid (11c C18:1)	1.19	0.02	1.14	0.03	1.14	0.02	0.12	0.35	0.23	1.00
TOT. MUFA	19.09	0.13	18.27	0.22	18.27	0.18	<0.001	0.01	0.001	1.00
Linoleic acid (C18:2)	14.28	0.13	14.30	0.21	13.71	0.17	0.02	1.00	0.03	0.09
DGLA (C20:3)	2.01	0.04	2.30	0.06	2.23	0.05	<0.001	<0.001	0.002	1.00
ARA (C20:4)	18.76	0.13	19.23	0.21	19.66	0.18	<0.001	0.18	<0.001	0.37
TOT. ω-6	35.06	0.15	35.83	0.25	35.65	0.21	0.12	0.03	0.08	1.00
EPA (C20:5)	0.60	0.02	0.49	0.03	0.54	0.03	0.01	0.02	0.35	0.66
DHA (C22:6)	4.97	0.11	4.67	0.17	4.79	0.14	0.29	0.41	0.95	1.00
TOT. ω-3	5.57	0.12	5.16	0.19	5.34	0.16	0.16	0.21	0.73	1.00
TOT. PUFA	40.63	0.14	40.99	0.22	40.98	0.18	0.21	0.51	0.39	1.00
Trans C18:1	0.08	0.01	0.09	0.01	0.09	0.01	0.88	1.00	1.00	1.00
>Trans C20:4	0.08	0.01	0.06	0.01	0.08	0.01	0.31	0.65	1.00	0.41
TOT. TRANS	0.17	0.01	0.15	0.01	0.17	0.01	0.47	0.71	1.00	0.72
Indexes										
ω-6/ω-3	6.59	0.18	7.33	0.28	7.23	0.24	0.09	0.09	0.10	1.00
Omega 3 Index	5.57	0.12	5.16	0.19	5.34	0.16	0.16	0.21	0.73	1.00
SFA/MUFA	2.12	0.02	2.21	0.03	2.24	0.02	<0.001	0.03	0.001	1.00
∆6D + ELO 20:3/18:2 ^b^	0.142	0.003	0.158	0.004	0.164	0.004	<0.001			
∆5D 20:4/20:3	9.59	0.18	8.46	0.29	8.96	0.24	0.004	0.004	0.12	0.59
∆9D 16:1/16:0	0.018	0.001	0.02	0.001	0.019	0.001	0.07	0.07	0.43	1
∆9D 18:1/18:0	0.994	0.01	0.928	0.017	0.916	0.014	<0.001	0.004	<0.001	1
PUFA BALANCE	13.71	0.28	12.58	0.45	13.00	0.37	0.08	0.11	0.40	1.00
Peroxidation Index	137.18	0.81	136.62	1.31	138.00	1.10	0.71	1.00	1.00	1.00
Unsaturation Index	161.58	0.57	161.27	0.91	162.28	0.76	0.67	1.00	1.00	1.00

Data are presented as mean ± standard error (SE). Adjusted for age, sex, and dietary components, extracted from the principal component analysis of dietary nutrient intake (individual FAs, families (SFA, MUFA, and PUFA), total lipids (%Energy), carbohydrates, fiber, proteins, and calories). ^a^ Post hoc tests were conducted with a Bonferroni adjustment. ^b^ Levene’s test of homogeneity of variance was not met.

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
