# Peer review of "Fatty Acid Profile of Mature Red Blood Cell Membranes and Dietary Intake as a New Approach to Characterize Children with Overweight and Obesity"

_nutrients, 2020, doi:10.3390/nu12113446_

Round 1

Reviewer 1 Report

Abstract: last sentence. I think this is too much. Maybe support the design…

Introduction: last sentence. The same as in the abstract. Is a personalized diet really necessary – stick to the rules which we already have and the plasma values will improve (your data show it, the weight status is the key factor!)

Methods/Results: I have big methodological concerns. You are comparing three groups. It is not acceptable to do paired tests between NO and OV; NO and OB; and OV and OB separately with Mann-Whitney U! Use the appropriate test and than apply for your many tests a p-value adjustment! It does not need to be Bonferroni which would be to strict. But using the false discovery rate (FDR) would be important (table 1, table 2, table 3)

Table 2 shows underreporting of OB and maybe OV, needs to be considered. Since whole intake of lipids % is according to your FFQ higher in NO than in OV the subanalysis (individual FA) is a result of this. Amount of whole lipids % needs to go into analysis for individual FA, e.g. ANCOVA.

The same for table 4: Please state in the methods more explicitly which variables were included where in your ANCOVA? Separately NO and OV; NO and OB; and OV and OB? If yes, not acceptable.  Across all your fatty acids I do not see where Bonferroni came in. Also here, adjusting your p-Values according to FDR is necessary.

Regarding

Discussion: First sentence: See comments for Introduction. Be a bit more modest. Reference 29 – what did they show. More context with relevant literature.

Last sentences: The data show that body weight appears to be the factor. Do not see the personalization yet from these data – FFQ is underreported. I recommend to stick to the data which remain significant after p-Value adjustments, to stay modest and to outline where the journey could go.

Reviewer 2 Report

This is an exploratory study that has provided some first basis information on the use of membrane lipid profiles in children with obesity.

The manuscript is well written, but there are some issues that need to be addressed to strengthen this report.

Major points

General

It is important to use ‘people first language’ throughout the manuscript. Sometimes it is used but more often it is not. Also for children with normal weight the term ‘normoweight’ is used. Suggest to just use the term ‘children with normal weight’.

Introduction

Aim: Is it not the main aim to determine whether or not these fatty acid profiles of red blood cell membranes can be a good biomarker of obesity? That is how I read it. The authors also state that using the methodology here a precise and personalized dietary strategy can be created. However, this study does not address this part of the aim in this manuscript. Furthermore, these are two completely different things: either looking for a good biomarker to determine obesity early in children or looking for how these fatty acid profiles can be a good biomarker to identify foods that should be changed in the diet. But in order to answer this latter part, a) these biomarkers should be good biomarkers for these foods, which is not explored very well in this manuscript (Dragsted et al, 2018, Genes Nutr 13:14) and b) should be explored whether changing diet through adding/eliminating these foods in the diet actually have an effect on these biomarkers. Only then you can say that this methodology can be used to create a personalized dietary strategy. Suggest to delete the personalized dietary strategy from the whole of this manuscript.

Material and methods

Obesity is classified using an age and sex-specific pediatric z-score using national Italian tables. However, is it not better to use international tables for this study that are standardized for international comparisons to make it more applicable for other countries? For example, the WHO or IOTF criteria could be used. Strongly suggest to rerun the analysis using these international growth charts. If not, the implications for the results and comparison with future studies on this topic should be fully explained in the Discussion section.

Paragraph 2.3, Food habits and nutrient intakes: the food frequency questionnaire was completed by the parents. Was this also the case for the adolescents, esp 14-16 yr olds? As older children have more control over their own food and drink intake and they might consume things their parents are not aware of. Please explain what an effect might this have had on the measurement of dietary intake.

Statistical analysis

The sample size is really small, especially the overweight group. Has a power calculation been done? Sometimes not seeing statistical significant differences might have to do with the groups being so small.

Results

Page 6, lines 198-199: hear the KIDMED index was compared between the group with normal weight children and the group with children with obesity, which showed a statistically significant difference. No significant difference was observed between the children with overweight and children with normal weight. However, the children with overweight had the same median and quartiles as the children with obesity. So, this is most probably due to power. On the other hand, though, the differences are really small, 1 median point on the index. The index ranges from 1-12 and is based on 16 questions. So, answering one question differently will have a huge effect on the calculated score. In the Discussion (page 9, first paragraph), the authors stated that the group with normal weight showed good adherence to the Mediterranean diet while the group with children with obesity and overweight showed mild adherence. Because of the small difference and if the cut-off was set at 9, it would have been a completely different story and no differences would have been found. This is a strong statement to make here. Suggest to change this in the Discussion section.

Page 6, lines 199 -202: the correlation coefficient mentioned might be significant statistically, but they are really weak, even for dietary correlations. This should be reflected in the interpretation, as it suggests now that the correlations are very important.

Paragraph 3.2: the ANCOVA analysis was adjusted for age, sex and dietary intake. Which dietary variables were included as that is not defined anywhere in the manuscript? Furthermore, suggest to adjust for SES as well and include general information on SES in the text (differences between children with normal weight, overweight and obesity with regard to sex). Social economic status has a huge effect on both the dietary intake and the obesity status of children and hence could drive some of the results. So, SES should be regarded and explored here as a confounder. If there is no data available for this, implications of this should be included in the Discussion section

Discussion

What is missing here, is a good discussion on good a biomarker the fatty acid profile of mature red blood cell membranes is for obesity. This study is just an explanatory study and this should be explained better. Also, what would future research have to do to explore this further and better.

Minor points

Abstract

It is stated that the aim of this study was to determine the fatty acid profile of red blood cell membranes as a comprehensive biomarker of children's obesity metabolism together with the evaluation of the nutritional status. However, in the manuscript the nutritional status is only determined by the dietary intake. The nutritional status is more than just dietary intake, it should include anthropometry too. But I have the feeling that the authors mean ‘dietary intake’ only here.

The final line states ‘a comprehensive membrane lipidomic profile of obesity will allow us to design precise dietary strategies to prevent and control obesity’. Suggest to finish the abstract with something based on the results or what future studies need to look at. As this statement will take a while before it becomes reality.

Material and methods

Is the study's design a cross sectional design? Please include which exact design is used for this study.

With regards to the consent procedure, did the children have to provide assent themselves to participate in the study?

Paragraph 2.2, Anthropometric measurements: it says here that body weight and height were measured by using standardized methods. Please include a reference here.

Table 3: suggest to merge this table with one of the other tables as it is only one line, one variable.

Round 2

Reviewer 2 Report

The manuscript has much improved due to the revisions, but there are a few final issues that should be addressed to strengthen this report even further.

General

It is important to use ‘people first language’ throughout the manuscript. This is not addressed yet. Instead of using ‘obese children’, write ‘children with obesity’ (see: http://www.uconnruddcenter.org/resources/upload/docs/what/bias/Putting%20People%20First%20in%20Obesity_Obesity_3.14.pdf)

Material and methods

Line 101: suggest to include ‘12-16 years of age’ instead of ‘12-16’.

With regards to use of national growth charts, sorry for the oversight with regards the tables that were used, I mixed things up. I agree with the authors with regards the controversies around the IOTF and WHO tables. However, if the results of this article are to be comparable to another study done on the same topic, it would be best to use similar methods, hence using internationally used growth tables and not national ones. If the authors decide to keep the results based on the national tables, at least address this in the Discussion section as a limitation of this study. And maybe some information on how comparable or how different the national growth charts are in comparison to IOTF and/or WHO.

With regards of filling out the FFQ, it is great that the authors included the underestimation of the intake as a limitation in the Discussion. Could the reference mentioned in the rebuttal be included in the text as well? I think it would also be informative for the reader to include in the Methods that the adolescents were encourage them to complete it themselves. Of course there is no way of checking this as it was done online, encouraging them is important.

Also, it was nowhere mentioned that the FFQ was filled out online, this should be included in the methods section.

Statistical analysis

Thanks for the elaborate answer with regards the sample size. Even though this is an exploratory study, a brief summary of the sample size calculation should be included in the manuscript. Also, the authors should include just in a sentence that there might not have been enough power to detect differences in the overweight group and could have led to a type II error. Even though this is an explanatory study and indeed the results are novel and hence deserve publication. Just acknowledging this limitation is still important.

Results

With regards the ANCOVA analysis, the authors say that the dietary variables used as covariates are explained more clearly now in the Methods section, but I was unable to find this information (line 182). Furthermore, please include this information as well in a footnote in table 3.

Discussion

The new section added really improves the Discussion and clearly explains why mature red blood cell membranes could be a good biomarker for obesity and how this study contributes to this.
